# Effects of Storage Duration and Structures on Sesame Seed Germination, Mold Growth, and Mycotoxin Accumulation

**DOI:** 10.3390/toxins15010039

**Published:** 2023-01-04

**Authors:** Samuel Alemayehu, Fetien Abay Abera, Kiros Meles Ayimut, Ross Darnell, Rizana Mahroof, Jagger Harvey, Bhadriraju Subramanyam

**Affiliations:** 1Department of Biology, College of Natural and Computational Sciences, Mekelle University, Mekelle P.O. Box 231, Tigray, Ethiopia; 2Department of Dryland Crop and Horticultural Sciences, College of Dryland Agriculture and Natural Resources, Mekelle University, Mekelle P.O. Box 231, Tigray, Ethiopia; 3Commonwealth Scientific and Industrial Research Organization, P.O. Box 2583, Brisbane 2601, Australia; 4Department of Biological and Physical Sciences, South Carolina State University, Orangeburg, SC 29117, USA; 5Feed the Future Innovation Lab for the Reduction of Post-Harvest Loss, Department of Plant Pathology, Kansas State University, Manhattan, KS 66506, USA; 6Department of Grain Science and Industry, Kansas State University, Manhattan, KS 66506, USA

**Keywords:** Ethiopia, fungal growth, germination, hermetic bags, moisture content, mycotoxin, sesame seed

## Abstract

Sesame is an important oil crop for the Ethiopian economy. However, the lack of adequate storage facilities results in significant losses of sesame seeds. This study was designed to compare the effects of storage conditions and the subsequent impact on sesame seed germination, mold growth, and mycotoxin accumulation over the storage period. The efficacy of two hermetic bags (1. Purdue Improved Crop Storage (PICS) bags and 2. Super GrainPro (SGP) bags) was directly compared to sesame storage in polypropylene (PP bags) and Jute bags. Storage conditions (oxygen, carbon dioxide, temperature, moisture content and relative humidity) of samples were analyzed in the laboratory in three replicates. Results showed that the oxygen concentrations dropped to 6.9% (±0.02) in PICS bags and 8.7% (±0.06) in SGP bags at the end of 6 months of storage. In non-hermetic bags Jute and PP), oxygen levels were close to atmospheric levels at 2-, 4-, and 6-month storage periods. In non-hermetic bags throughout storage, the amount of seed infection by mold constantly increased, and seed germination decreased. Sesame seeds stored in hermetic bags had 89.7% (±0.58) to 88.3% (±2.89) germination rates versus 61.67% (±2.08) for non-hermetic storage bags over the 6-month seed storage period. All mycotoxin levels increased over the same storage period, whereas comparative levels were much lower in hermetic bags after six months. Sesame seeds stored in both hermetic bags had the lowest level of tested mycotoxins, and levels among the SGP and PICS bags were not significantly different from one another. This study provides strong evidence indicating that hermetic storage structures such as PICS and SGP significantly affect temperature, humidity, moisture content, CO_2_ and oxygen levels resulting in the lowering of fungal growth and mycotoxin accumulation and effectively preserving stored sesame without relying on synthetic pesticides in Ethiopia.

## 1. Introduction

Post-harvest management of agricultural commodities is a significant challenge in developing countries, including Ethiopia [1]. In Ethiopia, 49% of grain crops, including oilseeds, are stored in traditional storage methods [2]. Dried commodities stored in such porous woven bags can gain moisture from environments with high relative humidity, allowing for fungal infections and the accumulation of mycotoxins [3,4]. Sesame is a widely cultivated oilseed crop and is the second leading Ethiopian export crop behind coffee [5]. Sesame accounts for over 90% of the value of oilseed exports and more than 30% of the total oilseed production [6,7]. Beyond the potential of production and increased export demand, sesame is susceptible to storage losses mostly due to insects and fungi [8,9]. The storage of sesame is mandatory because it is grown seasonally and stored as seeds for the next season as well as for sale when market prices are high. The period the seed can be safely stored depends on the conditions during harvesting, and the type of storage structures used [10]. Lack of adequate storage facilities can affect the quality of seeds and competitiveness in the export market. According to an FAO [10] report, 25–33% of the world grain crop, including seeds, is lost each year during storage which has a significant effect on global food security.

The presence of high oil content and fast cellular respiration of sesame seeds results in high grain moisture and storage temperature at levels that favor fungal growth [11]. Fungal activity in stored seeds can cause undesirable effects, including heating, seed discoloration, shrinking or damage, and losses in nutritional value and germination [12]. Certain seed-borne fungal pathogens produce mycotoxins, secondary metabolites that cause the deterioration of grain quality, a serious problem for seed producers, farmers, consumers, and exporters [13,14,15,16,17,18]. Additionally, mycotoxins lead to frequent rejections of export crops from import partners. Of biggest concern is the consumption of grains heavily contaminated with mycotoxin that can result in acute liver toxicity and cancers, is associated with stunting children’s development and immunosuppression, and acute poisoning can lead to human and animal death [19]. The patterns of granary temperature, relative humidity, grain moisture contents, level of gas concentration, and association of mold growth and mycotoxin accumulation with sesame seed stored in different methods are not documented in Ethiopia. Furthermore, due to fluctuations in international market prices and excess supply during peak harvesting seasons, farmers and traders frequently face a low market price for sesame. This negative impact can be mitigated by taking advantage of market prices when they are lower during the off -season.

Many post-harvest loss-mitigation technologies have proven effective [3], but the adoption of such technologies has been limited in most developing countries, including Ethiopia. Quantitative and qualitative loss of sesame seeds due to storage is a critical issue in Ethiopia, and improved storage technologies are needed to keep both the volume and quality of the seed for consumption, seeds for the next season and increase competitiveness in the export market. The use of hermetic storage bags, an alternative to pesticide use and conventional storage methods [20], can reduce post-harvest grain losses and lead to significant improvements in small-scale farmers’ food and nutrition security and to a positive economic impact. Hermetic storage bags prevent insect and mold infection of kernels, migration of moisture, diffusion of the high percentage of oxygen from ambient air, and preserve the germination capacity of the seeds [21,22]. Biological processes such as respiration and metabolic activities lead to the depletion of oxygen and the release of carbon dioxide inside hermetic containers [20]. Hermetic storage bags have been promoted in the Sub-Saharan region, including Ethiopia, to reduce post-harvest losses of maize, wheat, chickpea, and other vulnerable crops [3,21]. The effectiveness of hermetic bags in minimizing mold growth and accumulation of mycotoxin in stored chickpeas was demonstrated by Alemayehu et al. [3]. Yet, the use of PICS and SGP bags for storage of sesame seed by smallholder farmers is limited in Ethiopia. 

Studies conducted in different parts of the world so far were based on only maize and aflatoxin, a class of mycotoxins produced by Aspergillus fungi. Due to the increasing demand for high-quality seeds in the international market, providing basic information on key aspects of good storage technologies to farmers would help them practice appropriate management methods that maintain the quality and quantity of stored grain and seed. Studies on post-harvest technologies against fungal growth and mycotoxin accumulation and preserving the germination of sesame seeds have not previously been conducted in Ethiopia. Thus, there is a need to evaluate storage technologies that minimize mycotoxin contamination of sesame seeds and as well as preserve seed quality. This study was conducted to evaluate the effect of storage structures and storage durations on fungal growth, mycotoxin accumulation, and seed germination.

## 2. Results

### 2.1. Sesame Temperature and Moisture 

Table 1 shows the temperature, and moisture content (mean ± SE) recorded for the various storage bags tested during the storage periods. At the beginning of the storage experiment, there were no significant temperature differences between storage structures (F = 0.61; df = 3, 8; *p* = 0.62), whereas temperatures recorded at months 2, 4, and 6 were significantly different (F, range among months = 155.05–25,971.9; df = 3, 8; *p* < 0.001). The average temperature of sesame seeds stored in jute bags was the highest during the experiment, followed by PP bags, and levels in these two bags were significantly (*p* < 0.001) higher than in the hermetic bags. Throughout the storage periods, the temperature level of sesame seeds stored in PICS and SGP bags was the lowest, and the levels in these two bags were statistically similar at months 4 and 6.

Changes in the temperature and moisture content of sesame stored in different bags over successive periods are shown in Appendix A
Table A1. Temperatures decrease for all bag types at the 4–6 month period, increase during the 2–4 month period and decrease for all bag types’ bag types except for jute bag. The temperature in the jute bags increased significantly (*p* < 0.001) by an average of 0.25 °C each month, but there was no significant linear trend for polypropylene bags. As contrasted to this, the temperature in both hermetic bags decreased each month linearly by an average of 0.18 °C in PICS and 0.19 °C in SGP bags. Temperature levels in PICS bags fell from 20.15 ± 0.27(°C) in month 0 to 18.87 ± 0.06 (°C) in month 6. 

At the beginning of the storage experiment, there was no significant difference in sesame seed moisture content between bags (F = 1.23; df = 3, 8; *p* = 0.36), whereas moisture contents recorded at months 2, 4, and 6 were significantly different (F, range among months = 1653.16–24,021.84; df = 3, 8; *p* < 0.001). The moisture content of sesame seeds stored in jute bags was the highest throughout the storage periods, followed by PP bags, and levels in these two bags were significantly (*p* < 0.001) higher than in the hermetic bags. As shown by the line in Figure 1, the moisture content of sesame seeds increased in a linear trend for both jute and PP bags but remained constant in hermetic bags. The moisture content of sesame seeds stored in non-hermetic bags increased significantly (*p* < 0.001) by an average of 1.89% per month (Appendix A
Table A1). Following that, the moisture content of sesame seeds increased by 1.92% per month in Jute bags and 1.86% in PP bags, with no significant change in PICS and SGP bags. The relative humidity of sesame seeds stored in jute bags was also the highest throughout the storage periods, followed by PP bags, and levels in these two bags were significantly (*p* < 0.001) higher than in the hermetic bags. 

### 2.2. Oxygen and Carbon Dioxide Concentrations in Storage Bags

Table 1 shows the oxygen and carbon dioxide concentrations (mean ± SE) recorded for the various storage bags over the storage periods. At the start of the storage experiment, there were no significant differences in oxygen (F = 0.01; df = 3, 8; *p* = 1.00) or carbon dioxide concentrations (F = 0.16; df = 3,8; *p* = 0.92) between the storage bags, but the concentrations between storage structures recorded at 2, 4, and 6 months were significantly different (*p* < 0.001). At month 2, oxygen concentrations in both traditional bags were higher than in hermetic bags, and levels within bags were statistically similar. At months 4 and 6, sesame seeds stored in Jute bags had the highest oxygen concentrations, followed by PP bags, and levels in these two bags were significantly (*p* < 0.001) higher than in the hermetic bags. However, the oxygen concentrations in both hermetically sealed bags changed during storage. As a result, after 6 months of storage, the oxygen level in PICS bags dropped to 6.87% and 8.67% in SGP bags, with the levels in PICS bags significantly lower than those in SGP bags. Figure 1 shows the trend in oxygen reduction and carbon dioxide increase after 2, 4, and 6 storage periods for hermetic and non-hermetic bags. Changes in oxygen levels between hermetic and non-hermetic bags were minor for the first two months but significant (*p* < 0.001) for the next 4-6 months period (Appendix A
Table A1). Oxygen levels in hermetic bags dropped by 11% compared to 0.73% in non-hermetic bags. Between 0–2, 4–2, and 4–6 months, oxygen concentrations in PICS bags decreased by 0.73, 0.45, and 12.04%, respectively, and in SGP by 0.46, 1.14, and 9.8%, but changes between observation months were not significant in non-hermetic bags.

The carbon dioxide levels in all bags increased by different amounts during the trial. At months 2 and 6, carbon dioxide concentrations were highest in SGP bags (1.15 ± 0.03%), followed by PICS bags (1.05 ± 0.02%), and lowest in both traditional bags. Appendix A
Table A2 shows there was no significant change in carbon dioxide concentrations for the first two months, significantly (*p* < 0.05) increased by 17.03% in SGP bags and 10.70% in PICS bags between months 4 and 6. The carbon dioxide increase in the non-hermetic bags was small and not significant (<0.6% per period), while the level in the hermetic bags was much larger, with the largest increase of 13.9% occurring in the 4–6 month period.

### 2.3. Mold Infection and Germination of Sesame Seeds

The percentage of mold infection and germination (mean±SE) of sesame seeds stored in the four storage structures over the storage periods is shown in Table 2. There was no significant difference in the levels of mold infection between the storage structures at the beginning of the storage experiment (F = 0.08, df = 3, 8 and *p* = 0.97), but at months 2, 4, and 6, there were significantly different (F, range among months = 155.05–25,971.9; df = 3, 8; *p* < 0.001). After 6 months of storage, the percentage of sesame seeds infected by mold decreased from 38.33% to 23.67% in the PICS bags and from 38.47% to 22.67% in the SGP bags compared to 0 months. Sesame seeds stored in hermetic bags had the lowest mold infection, and levels were statistically similar after 6 months of storage. Between 0–2 months, the percentage of sesame seeds infected with mold increased by 23.3% in jute bags and 9.32% in PP bags but decreased by 14.7% in PICS bags and 15.8% in SGP bags at the end of storage periods.

Table 2 shows that at the start of the storage experiment, there was no significant difference in sesame seed germination between storage structures (F = 1.00, df = 3, 8 and *p* = 0.44), but seed germination recorded at months 2, 4, and 6 were significantly different (F, range among months = 54.37–1951.5; df = 3, 8; *p* < 0.001). After 6 months of storage, compared to 0 months, sesame seed germination in PP bags decreased from 90.00% to 61.67% and from 89.97% to 61.67% in jute bags. At the end of storage periods, seed germination decreased in both traditional bags by 28.33% but was negligibly decreased in PICS and SGP bags. Despite this, there is no significant difference in seed germination over the storage periods between hermetic storage bags. 

### 2.4. Fungal Infection of Stored Sesame Seed

Table 3 shows the percentage of sesame seeds infected by *Aspergillus, Penicillium* and *Fusarium* spp. in different storage structures. At months 2 and 6, the percentage of sesame seeds infected by *Aspergillus* spp. was significantly higher in the Jute and PP bags than in the PICS and SGP bags. Even though the proportion of seeds with *Aspergillus* infection in the PICS bag was significantly higher than that in the SGP bag on month 2, it was statistically similar in months 4 and 6. While the percentage of *Aspergillus* species-infected seeds was statistically similar in the PP bag and the Jute bag in month 4, there were more infected seeds in the Jute bag than in the PP bags in months 2 and 6. After 2, 4, and 6 months of storage, the percentage of sesame seeds infected by *Aspergillus* spp. in traditional bags increased steadily, with 35.2%, 43.7%, and 47.1% in PP bags and 39.5%, 43.7%, and 49.4% in jute bags, respectively. During the six-month storage period, the percentage of seeds infected by *Aspergillus* in hermetic bags decreased from 28.0 ± 0.8% to 16.9% ± 0.4 in the PICS bags and from 29.5 ± 0.8 to 18.4 ± 1.1% in the SGP bags. The percentages of seeds infected by *Penicillium* spp. were significantly higher in the Jute and PP bags throughout storage periods than those of the PICS and SGP bags. Overall, the percentage of seeds infected by *Penicillium* spp. increased by 14.82% in Jute bags and 14.75% in PP bags after 6 months of storage. While the percentage of seeds with *Penicillium* spp. infection decreased over the course of the six-month storage periods, progressing from 20.4% to 15.3% in PICS bags and 17.7% in SGP bags. Throughout storage periods, the percentages of seeds with *Fusarium* spp. infection decreased in all storage structures, but they fell by 14.07% in PICS bags and SGP bags.

### 2.5. Mycotoxin Levels of Stored Sesame Seeds

The levels of total aflatoxin, total fumonisins, deoxynivalenol, and ochratoxin (mean ± SE) in sesame seeds stored in different storage structures are shown in Table 4. The levels of each mycotoxin are discussed separately below.

#### 2.5.1. Total Aflatoxin

At the beginning of the storage experiment, the aflatoxin content of sesame seeds was not significantly different among the four storage structures (F = 0.88; df = 3, 8; *p* = 0.49), but levels detected at months 2, 4, and 6 were significantly different (F, range among months = 1065–296,492; df = 3, 8; *p* < 0.001). Sesame seeds stored in PP bags (6.91 ± 0.02 ppb) had the highest levels of aflatoxin at month 2, followed by jute bags (5.95 ± 0.03 ppb). At months 4 and 6, total aflatoxin levels in jute bags were significantly higher (*p* < 0.001) than in the rest of the storage bags. In contrast to this, sesame stored in hermetic bags had the lowest overall aflatoxin levels, and the levels in these two storage bags were not significantly different at the end of storage.

Appendix ATable A3 show the change in aflatoxin levels of sesame seeds for each period for each of the treatments. The aflatoxin content of sesame stored in all storage bags increased between months 0 and 2, with increases being significantly greater in traditional PP bags. Between months 2 and 4, the average aflatoxin level of sesame seeds increased in jute and polypropylene bags by 8.08 and 6.00 ppb, respectively, while its increment ranged from 0.01 to 0.56 ppb in hermetic bags. Similarly, between months 4 and 6, aflatoxin levels of sesame seeds in jute and polypropylene bags increased by 2.10 and 2.82 ppb, respectively, whereas the change in both hermetic bags ranged from 0.04 to 0.21 ppb. Throughout storage periods, the overall change in aflatoxin levels of sesame seeds stored in hermetic bags ranged from 0.13 to 1.20 ppb. 

#### 2.5.2. Total Fumonisin

The mean total fumonisin levels of sesame seeds in the four storage structures at month 0 were 0.4 ± 0.0 ppm. The fumonisin levels of sesame seeds stored in different storage structures were not significantly different at month 0 (F = 1.108; df = 3, 8; *p* = 0.401), whereas they varied significantly among storage structures at months 2, 4 and 6 (F, range among months = 120,013–675,680; df = 3, 8; *p* < 0.001). At months 2, 4, and 6, the highest fumonisin levels of sesame seeds were detected in jute bags, followed by SGP bags, and levels between these two storage bags were significantly different (*p* < 0.001). Total fumonisin levels in sesame seeds remained constant in both hermetic bags throughout the storage period. Between months 0 and 2, fumonisin levels of sesame seeds in jute and polypropylene bags increased by 1.17 and 0.83 ppm, respectively. Overall, fumonisin levels of sesame seeds in PP bags increased by 0.6 and 0.14 ppm between 2–4 and 4–6, respectively. Between months 4 and 6, fumonisin levels in jute bags increased by 0.26 ppm.

#### 2.5.3. Deoxynivalenol

One-way ANOVA revealed that deoxynivalenol levels of sesame seeds stored in all four storage structures were similar at month 0 (F = 1.33; df = 3, 8; *p* = 0.33), but significantly different at months 2, 4, and 6 (F, range among months = 105,480–144,134; df = 3,8; *p* < 0.001). At months 2, 4, and 6, the highest deoxynivalenol level of sesame seeds was detected in jute bags, followed by PP bags, and levels between these two traditional storage bags were significantly (*p* < 0.05) different. Between 0 and 2 months of storage, there was a progressive increase in the deoxynivalenol content of sesame in all storage bags. Deoxynivalenol levels of sesame seeds in jute and polypropylene bags increased by 0.97 and 0.34 ppb, respectively, between months 2 and 4. Between months 4 and 6, deoxynivalenol levels in sesame seeds increased by 0.69 and 0.21 ppb in jute and polypropylene bags, respectively. Deoxynivalenol levels of seeds stored in PICS and SGP bags remained unchanged at months 2, 4, and 6. 

#### 2.5.4. Ochratoxin A

The ochratoxin levels of sesame seeds across storage structures at month 0 were not significantly different (F = 0.46; df = 3, 8; *p* = 0.72), whereas levels varied significantly among storage structures at months 2, 4, and 6 (F, range among months = 862.6–2140; df = 3, 8; *p* < 0.001). Ochratoxin levels were the highest for sesame seeds stored in jute bags, followed by PP bags throughout the storage period and levels among these two traditional storage bags were significantly (*p* < 0.05) different from one another. Ochratoxin levels were similar for sesame seeds stored in PICS and SGP bags at months 2 and 4. At month 6, ochratoxin levels of sesame seeds stored in PICS and SGP bags were not significantly different (*p* < 0.05) among these two hermetic storage bags (Table 5). Changes in Ochratoxin A were not significant in hermetic bags between 0-2 and 4-6 months but significant in non-hermetic bags throughout the observation period (Appendix A
Table A3).

### 2.6. The Correlation of Fungal Contamination and Mycotoxin Levels

*Aspergillus*, and *Penicillium*, were positively and significantly (*p* < 0.05) correlated with all toxin productions (Table 5). *Aspergillus* spp., had positive correlations with aflatoxin content: *Aspergillus* spp. (r = 0.94, *p* < 0.001), level of total aflatoxin (r = 0.91), total fumonisins (r = 0.91), DON (r = 0.90) and Ochratoxin (r = 0.86). Unlike this, *Fusarium* spp. show negative and significant (*p* < 0.05) correlation with all toxin productions. The coexistence of *Penicillium* and *Aspergillus* spp. has a significant impact on ochratoxin A in sesame stored in conventional structures.

## 3. Discussion

### 3.1. Sesame Seed Moisture and Temperature

Seeds and grains can be stored in a variety of structures under various storage conditions as food reserves, seeds for the next season, and to increase marketability [2]. Temperature, relative humidity, and seed moisture content are all important climatic factors for maintaining and preserving healthy seeds during storage [3,14,20]. As the storage period increased, the moisture content of sesame seeds increased significantly (*p* < 0.001) by 1.92% in Jute bags and 1.86% in PP bags per month but remained constant in PICS and SGP bags throughout the storage periods. The porous nature of traditional storage structures, which allowed the seeds to absorb moisture from the environment, could contribute to the increase in the moisture content of seeds. This was consistent with Mendoza et al. [23] and Walker et al. [24], who found that the moisture content of maize increased significantly during conventional storage. However, both PICS and SGP bags maintained sesame seed moisture within the safe storage range of 6-8% throughout the storage period [25], indicating a lack of exchanges between hermetic bags and the outside environment. These findings are consistent with the findings of various studies that have demonstrated the efficacy of hermetic bags [26,27]. The temperature in jute bags increased significantly (*p* < 0.001) by 0.25 °C on average each month, but the temperature in both hermetic bags decreased each month linearly by 0.18 °C in PICS and 0.19 °C in SGP bags. Jute bags’ high moisture content and constant oxygen supply promote fungal growth, which generates heat during respiration. Lane and Woloshuk [28] also indicated that fungal and insect respiration could cause hot spots within the stored grain.

### 3.2. Oxygen and Carbon Dioxide Concentrations in Storage Bags

Our results confirmed a significant decrease in oxygen concentration and an increase in CO_2_ level in both hermetic bags when compared to non-hermetic bags. This finding is consistent with the findings of Alemayehu et al., Borisjuk and Rolletschek [3,29], who confirmed that insect, fungi, and grain respiration depletes oxygen and releases carbon dioxide inside hermetic containers. Contrary to previous work, oxygen concentration decreased in PICS bags and SGP bags slowly at months 2 and 4 might be due to low initial moisture content, which contributed to low metabolic activities of seeds and fungi in the storage bags [26]. Williams et al. [21] found that the oxygen concentration of maize stored at low moisture levels in PICS bags remained above 10% after two months, but at higher moisture, the oxygen concentration was near zero after one month. The other two non-hermetic storage bags, on the other hand, did not significantly contribute to oxygen depletion, and the evolution of carbon dioxide and level were close to atmospheric levels throughout all assessment periods. This could be due to the bags’ lack of hermetic conditions, which likely facilitated gaseous exchange between the outside environment and the inside of the bags during storage [24,25,26,27,28]. 

### 3.3. Mold Infection and Germination of Sesame Seeds

Storage technologies that can prevent moisture migration, oxygen diffusion, and mold growth are required to reduce quantitative and qualitative loss of stored products. During the storage experiment, infection s of sesame seeds by mold was reduced by 7.0 to 8.5% in hermetic bags per month between months 4 and 6 but increased by 12.3% to 16.32% in non-hermetic bags. The rapid increase in mold infection of sesame seeds over storage periods may be related to the rapid increase in moisture, temperature, and oxygen supply in non-hermetic bags [26]. Similar investigations have reported that hermetic storage is significantly superior to non-hermetic storage in reducing fungal infection and preservation of seed germination during storage [3,8]. Mold infection has been linked to a persistent decrease in seed germination in non-hermetic bags because they inhibit amino acid integration into proteins, preventing embryo germination [30]. Sesame seeds stored in both traditional bags showed a progressive increase in *Aspergillus* and *Penicillium* spp. infection, but a decline in *Fusarium* spp. Warm temperatures and high moisture content, combined with high-level gas exchange, are likely to create favorable conditions for fungal growth and mycotoxin production in sesame stored in traditional storage in Ethiopia. The significant differences observed were explained by the storage structure and storage duration effects.

Although the storage bags tested were designed for grain storage rather than seed storage, Ethiopian farmers and agricultural extension agents have been persistent in their attempt to determine whether hermetic storage techniques for grain could also be used for seed storage. The current study provides strong confirmation demonstrating that both hermetic bags maintained seed germination within the safe storage range of 85–90% throughout storage periods [31]. Seeds stored in non-hermetic bags, on the other hand, had a 28.3% reduction in germination after 6 months, compared to less than a 2% reduction in other hermetic bags. These storage losses harm small-scale farmers’ livelihoods by increasing food insecurity and decreasing household income. This is also consistent with the findings of Njoroge et al. [32], who discovered that germination rates of maize decreased from 91% to 36% after 6 months of storage in polypropylene bags. The slight decrease in seed germination observed in the hermetically stored bags could primarily be due to the quality of the initial seeds.

### 3.4. Mycotoxin Levels of Stored Sesame Seeds

Mold and mycotoxin have a positive impact on food safety, international trade, and health in developing countries [33]. Hermetic storage significantly reduced the increase in mycotoxins compared to non-hermetic bags by maintaining storage conditions of stored seeds. Our current findings show a positive and significant relationship between mycotoxin levels and moisture content. The cumulative effect of rising temperatures, moisture, relative humidity, and oxygen supply in seeds stored in non-hermetic bags most likely contributed to increased fungal growth and mycotoxin buildup [24,25,26]. Increased infection by *Aspergillus* spp. contributed significantly to the rapid rise in aflatoxin levels in sesame seeds stored in traditional bags. Aflatoxin accumulation in traditional bags was investigated using correlation analyses as a function of *Aspergillus* spp. In traditional bags, all had positive correlations with aflatoxin content. Our findings were consistent with those of Ahmad and Singh [34], who found maize stored in Jute bags in India was associated with an increase in moisture content and high aflatoxin contamination. It would be unsafe to store sesame seeds in non-hermetic bags due to increased mycotoxin development, which might result in associated health issues and an international market barrier. Among the storage structures tested, PICS and SGP bags maintained a safe level of total aflatoxin (<20 ppb FDA and <10 ppb East African standards) throughout the six-month storage period. Our findings show that increased moisture, temperature, humidity, and infection of sesame seeds with *Aspergillus* spp. all significantly influenced the aflatoxin production in traditional bags. The rapid increase in ochratoxin levels in sesame seeds stored in traditional bags was also significantly influenced by the combination of *Aspergillus* and *Penicillium* spp. infection. A positive relationship was discovered between ochratoxin content and *Aspergillus* and *Penicillium* spp. in traditional bags. The current studies allowed us to successfully test different storage bags for the preservation of sesame seeds in a laboratory setting.

## 4. Conclusions

Sesame is Ethiopia’s most important oil crop in terms of income, food, and nutritional supply. In order to meet the rising demand for high-quality foods and sesame export standards, improved seed preservation is urgently needed. Infection of sesame seeds by storage mold lowers seed quality and increases mycotoxin production. The current study compared the effects of hermetic and non-hermetic sesame storage bags on physical environments, fungal infection, mycotoxin accumulation, and seed germination over a six-month period. When compared to non-hermetic storage bags, PICS and SGP bags were able to maintain sesame seed moisture content, intergranular temperature, relative humidity, and oxygen concentration. This indicates that storing sesame seeds in hermetic bags reduces the negative effects of storage conditions and can promote fungal storage growth and mycotoxin accumulation. Both hermetic bags act as a barrier to moisture migration and gaseous exchange between the outside and inside of the bags during storage, which can contribute to storage fungal growth and seed/grain infection. The findings revealed that hermetic storage in PICS and SGP bags reduced mold infection and mycotoxin accumulation during storage. The current study provides strong evidence that both hermetic bags maintained seed germination within the safe storage range of 85–90%. Whereas sesame seeds stored in non-hermetic bags demonstrated increases in mold infection and decreases in seed germination throughout the storage period. The levels of total aflatoxin, total fumonisins, deoxynivalenol, and ochratoxin increased significantly in conventional storage bags. The measured parameter in hermetic and non-hermetic storage bags clearly showed that hermetic storage, such as in PICS and SGP bags, is the preferred option for storing sesame. This result recommends that using these storage hermetic bags allows users to maintain the quality of the seed with less storage loss for extended periods of time, offering producers and traders the opportunity to fetch better market prices and supply healthy food to consumers.

## 5. Materials and Methods

### 5.1. Treatments and Experimental Design

The experiment was conducted at Mekelle University in Ethiopia for 6 months (July–December 2017). Two 100 kg capacity sealed bags (PICS bag and SGP bag) and the traditional 100 kg capacity polypropylene (PP) and jute storage bags have been tested using three duplicate fully random designs [3]. The hermetic bags were obtained from Shayashone Trading PLC and HiTEC Trading PLC in Addis Ababa, Ethiopia, respectively. The PICS bags contain three layers, while the SGP bags have only one layer of 78 μm polyethylene plastic film. Two traditional bags were purchased at a local market in Mekelle, Ethiopia. All sealed storage bags are checked for leaks before packing sesame seeds.

### 5.2. Sesames Seed Sources

Sesame seeds commonly grown in Ethiopia (Setit I) were purchased from the Seed Production and Distribution Company (PVT), Humera. The seeds used for the study were harvested in 2016 and were not treated with any pesticides. Seeds after purchase were stored at room temperature until used for experiments. The seeds were examined for the presence of mycotoxigenic fungi and for toxicity levels. Tests showed the seeds were naturally contaminated with mycotoxins, but initial toxin levels were much lower than the maximum allowable limit for each of the four mycotoxins tested. All replicates started from the same cohort of infected samples [3] and were carefully mixed. The homogenized seeds were divided into 50 kg lots and were attributed to every four storage bags: PICS bags, SGP bags, PP bags, and jute bags. PICS and SGP bags act as hermetic storage technologies, while PP and jute bags represent traditional storage materials used by small farmers in Ethiopia. Each of the storage bags was filled with 50 kg of sesame seeds and sealed using the twist-tie method, following the manufacturer’s instructions, with headspace above the seeds in each bag squeezed well to remove excess air and stored at ambient room conditions. For each storage structure, a kilogram of sesame seeds is performed when the experiments (0) and after 2, 4 and 6 months of storage. Each storage structure and sampling month combination is repeated three times, and each storage is sampled over time.

### 5.3. Storage Environment and Sesame Sampling Protocol

The oxygen and carbon dioxide levels in each storage bag were measured as described by Kalsa et al. [20]. After determining the gas measurements, each copy of the storage structure was opened to obtain 1 kg of the sample, which was collected by sampling from the upper, middle and lower parts of each structure using a handheld sampler, according to the guidelines of the International Seed Testing [35]. All sesame samples were stored at 4 °C in a refrigerator until fungal isolation and germination testing were completed, then samples were stored at −20 °C for mycotoxin analysis. The temperatures of sesame seeds and the equilibrium moisture content of each bag were also measured using an inexpensive meter [4]. This meter utilizes temperature and relative humidity measurements to predict equilibrium moisture content, and this meter has been validated using black sesame. These measurements have been done over 1 kg of sesame samples taken from each replicate of the storage structure after checking the oxygen level and carbon dioxide first in each storage structure.

### 5.4. Mold Infection

Sesame seeds collected from each storage bag were surface sterilized in 1% sodium hypochlorite solution for one minute and rinsed seeds three times using sterile distilled water [22]. From each replication of a storage structure, 100 surface-sterilized sesame seeds were plated on potato extract-dextrose agar (PDA) medium, a total of 25 seeds per 12 mm diameter PDA plate [36]. Plates were stored in a multi-room incubator (WITEG Labortechnik, Wertheim, Germany) at 28 °C and 60–70% r.h. recording of any fungal growth in each plate and seed began after seven days of incubation. The number of infected seeds in each plate was counted to determine the percentage of infected seeds. The frequency of fungi was calculated as the number of seeds containing a particular fungus × 100/total number of seeds used. The fungi from each of the infected seeds in every plate were isolated and were transferred and maintained on agar slants to determine the fungal species later.

### 5.5. Seed Germination

The seed germination test was undertaken to know if hermetically stored sesame would provide the benefit of viable seeds in addition to grain preservation for food purposes. For the measurement of germination capacity, three replicates of 100 randomly selected sesame seeds from each storage bag were placed in a pair of 215 × 215 mm plastic Petri dishes (50 seeds per plate) lined with filter papers. The filter papers were moistened with sterile distilled water to facilitate germination [36]. The seeds in each Petri dish were placed approximately at equal distances from each other. The plates were kept in a germination chamber fitted with fluorescent lights, with a 12 h light and 12 h dark-light cycle, at a temperature of 25 °C. Sterile distilled water was added to each plate on a daily basis to maintain adequate moisture in each plate. Records of any germination capacity in each plate and seed commenced after seven days of incubation. The number of germinated seeds in each plate was counted to determine the seed germination capacity. The germination percentage was calculated by the number of seeds germinated × 100/total seeds used.

### 5.6. Mycotoxin Analysis

From each 1 kg of the sample taken from each bag at each replicate, 300 g of sesame seeds was used for mycotoxin analysis. Before mycotoxin analysis, samples were ground to a fine powder using an electric blender (Model: DE500g, China), sieved with a 1 mm mesh sieve, and stored in a sealed Ziplock bag at −20 °C. Sample preparation, extraction and mycotoxin analysis were performed according to the manufacturer’s instructions [37]. Total aflatoxins, total fumonisins, and deoxynivalenol levels were determined using the AgraStrip watex test kits, COKAS1600WS, COKAS3000A, and COKAS4000A, respectively, and quantified using a lateral flow AgraVision reader developed by Romer Labs^®^ Inc. (Union, MI, USA). Ochratoxin A (OTA) levels were determined using a Stat Fax 4700 Enzyme-Linked Immuno-Sorbent Assay reader (Romer Labs^®^ Inc., Union, MI, USA) [37,38,39,40]. The test was carried out in accordance with the manufacturer’s instructions and has previously been published [3,23]. All test kits used to determine toxin levels in this study were USDAGIPSA-approved and purchased from Romer Laboratories (Union, MI, USA). Mycotoxin test kits have been validated by Romer Labs scientists with sesame samples with known levels of mycotoxins. Quantitative results were recorded in parts per billion and parts per million as specified by the manufacturer of test kits. The range of quantification (ROQ) for aflatoxin was 0–100 ppb, with a limit of detection (LOD) of 3.31 ppb and a limit of quantitation (LOQ) of 5 ppb, while the ROQ for fumonisin is 0–5000 ppb, with a LOD of 300 ppb and a LOQ of 400 ppb. The ROQ for deoxynivalenol is 0–5000 ppb, with a LOD of 210 ppb and a LOQ of 250 ppb and that of ochratoxin A 2–40 ppb, with a LOD of 1.9 ppb and a LOQ of 2 ppb [23,39]. For readings above the upper limit, the extract was diluted until a measurement within the detection range was obtained, and the final concentration was recorded after multiplying by the appropriate dilution factor(s).

### 5.7. Data Analysis

Data on temperature, humidity, oxygen and carbon dioxide concentrations of preserved sesame seeds, percentage of seed infection and germination, and levels of each of the four mycotoxins recorded in each bag were first subjected to analysis of variance (ANOVA) to determine the significant difference between the storage structures, the month of sampling, and the interaction between the storage structure and the month of sampling [41]. Data were then sampled by one-way ANOVA each month to determine significant differences between storage structures for each variable studied. If the one-way ANOVA was significant (*p* < 0.05), means among storage structures were separated using Tukey’s Honest Significant Difference (HSD) test [41]. Data of all variables were analyzed without any transformation. Thus, all means and standard deviations presented in all tables were based on untransformed data.

## Figures and Tables

**Figure 1 toxins-15-00039-f001:**
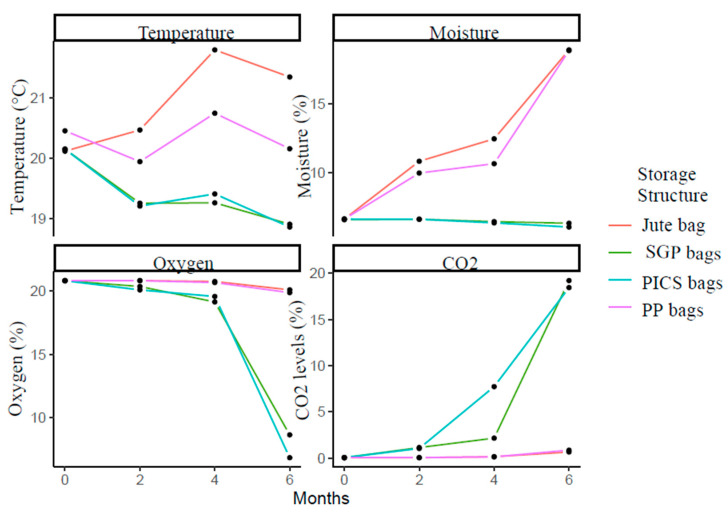
Temperature, moisture, oxygen, and carbon dioxide levels of sesame are stored in different storage structures over storage months. The line represents the means of the variables for each treatment.

**Table 1 toxins-15-00039-t001:** Mean ± SE temperature, moisture, relative humidity, oxygen, and carbon dioxide levels of sesame stored in different storage structures at 0, 2, 4 and 6 months. Means with the same letter are not significantly different.

Variables	Storage Structure	Mean ± SE at Month			
		0	2	4	6
Temperature (°C)	PICS bags	20.15 ± 0.27	19.21 ± 0.01d	19.41 ± 0.01c	18.87 ± 0.06c
	SGP bags	20.15 ± 0.27	19.26 ± 0.00c	19.27 ± 0.01c	18.91 ± 0.02c
	PP bags	20.46 ± 0.01	19.95 ± 0.01b	20.75 ± 0.27b	20.16 ± 0.02b
	Jute bags	20.12 ± 0.59	20.47 ± 0.01a	21.79 ± 0.02a	21.35 ± 0.32a
Moisture (%)	PICS bags	6.64 ± 0.02	6.63 ± 0.02c	6.37 ± 0.01d	6.07 ± 0.01d
	SGP bags	6.59 ± 0.06	6.63 ± 0.03c	6.46 ± 0.01c	6.34 ± 0.00c
	PP bags	6.64 ± 0.02	9.97 ± 0.04b	10.65 ± 0.02b	18.83 ± 0.01b
	Jute bags	6.64 ± 0.02	10.82 ± 0.18a	12.45 ± 0.02a	18.88 ± 0.01a
Humidity (%)	PICS bags	58.37 ± 2.55	44.88 ± 0.16c	40.06 ± 0.03c	38.75 ± 0.57d
	SGP bags	58.38 ± 2.55	38.23 ± 0.21d	37.86 ± 0.76d	46.24 ± 0.03c
	PP bags	58.64 ± 2.33	57.97 ± 0.09b	60.94 ± 0.15b	62.46 ± 0.04b
	Jute bags	58.37 ± 2.55	59.96 ± 0.20a	70.63 ± 0.78a	74.61 ± 0.01a
Oxygen (%)	PICS bags	20.81 ± 0.04	20.08 ± 0.01b	19.56 ± 0.03c	6.87 ± 0.02d
	SGP bags	20.81 ± 0.04	20.35 ± 0.23b	19.13 ± 0.01d	8.67 ± 0.06c
	PP bags	20.81 ± 0.04	20.81 ± 0.04a	20.66 ± 0.02b	19.87 ± 0.06b
	Jute bags	20.81 ± 0.04	20.82 ± 0.02a	20.74 ± 0.04a	20.09 ± 0.01a
CO_2_ levels (%)	PICS bags	0.06 ± 0.03	1.05 ± 0.02b	7.74 ± 0.05a	18.44 ± 0.02b
	SGP bags	0.06 ± 0.03	1.15 ± 0.03a	2.17 ± 0.01b	19.20 ± 0.10a
	PP bags	0.07 ± 0.03	0.07 ± 0.03c	0.15 ± 0.04c	0.87 ± 0.01c
	Jute bags	0.06 ± 0.03	0.05 ± 0.01c	0.16 ± 0.01c	0.67 ± 0.02d

Each mean is based on n = 3 replications. SE that is 0.0 falls in the range of 0.0 to 0.003. Means for each storage condition among storage structures by month followed by different letters are significantly different (*p* < 0.05; by Tukey’s HSD test).

**Table 2 toxins-15-00039-t002:** Mean ± SE Mold infection and germination levels (%) of sesame seeds stored in different storage structures at 0, 2, 4 and 6 months. Means with the same letter are not significantly different.

Variable	Storage Structure	Month 0	Month 2	Month 4	Month 6
Seed germination	PICS bags	90.00 ± 0.00	89.00 ± 1.73a	88.33 ± 2.89a	89.67 ± 0.58a
	SGP bags	90.00 ± 0.00	89.67 ± 0.58a	88.33 ± 2.89a	88.33 ± 2.89a
	PP bags	90.00 ± 0.00	77.67 ± 2.52b	69.67 ± 0.58b	61.67 ± 2.08b
	Jute bags	89.97 ± 0.06	72.00 ± 2.65c	67.33 ± 1.15b	61.67 ± 1.53b
Mold Infection	PICS bags	38.33 ± 0.12	38.33 ± 1.53c	33.33 ± 2.08b	23.67 ± 0.58b
	SGP bags	38.47 ± 0.15	39.00 ± 1.00c	30.00 ± 1.00c	22.67 ± 0.58b
	PP bags	38.01 ± 0.05	47.33 ± 0.58b	66.33 ± 0.58a	79.67 ± 1.53a
	Jute bags	38.33 ± 0.06	61.67 ± 2.08a	67.33 ± 0.58a	78.67 ± 1.53a

Means for each storage condition among storage structures by month followed by different letters are significantly different (*p* < 0.05; by Tukey’s HSD test).

**Table 3 toxins-15-00039-t003:** Mean ± SE the percentage of sesame seeds infected by *Aspergillus, Penicillium* and *Fusarium* spp. in different storage structures at 0, 2, 4 and 6 months.

Fungal Genera (%)	Storage Structure	Month 0	Month 2	Month 4	Month 6
*Aspergillus*	PICS bags	28.03 ± 0.58	20.90 ± 0.82c	20.00 ± 0.60b	16.93± 0.40c
	SGP bags	29.53 ± 0.81	17.93 ± 0.45d	20.03 ± 0.65b	18.43± 1.17c
	PPbags	29.53 ± 0.81	35.20 ± 1.11b	43.70 ± 0.35a	47.13±0.67b
	Jute bags	28.80 ± 1.11	39.47 ± 0.47a	44.57 ± 0.40a	49.40± 0.60a
*Fusarium*	PICS bags	32.87 ± 4.07	26.07 ± 2.48b	16.96 ± 0.79c	14.07± 0.40c
	SGP bags	33.63 ± 0.58	26.50 ± 2.10b	19.98 ± 0.63b	14.07± 0.40c
	PP bags	33.63 ± 0.58	28.90 ± 1.56b	26.67 ± 0.78a	18.83± 0.35b
	Jute bags	33.60 ± 2.36	44.37 ± 0.60a	26.27 ± 0.78a	22.00±1.66a
*Penicillium*	PICS bags	20.40 ± 0.46	21.32 ± 0.28c	18.98 ± 1.24c	15.27± 0.46c
	SGP bags	20.40 ± 0.46	21.37 ± 0.31c	20.03 ± 0.65c	17.67± 0.46b
	PP bags	20.01 ± 0.02	25.45 ± 0.13b	28.67 ± 0.23b	34.76± 0.21a
	Jute bags	20.47 ± 0.38	26.45 ± 0.22a	34.15 ± 0.29a	35.29± 0.25a

Means with the same letter are not significantly different (*p* < 0.05; by Tukey’s HSD test).

**Table 4 toxins-15-00039-t004:** Mean ± SE mycotoxin levels of sesame seeds stored in different storage structures at 0, 2, 4 and 6 months.

Mycotoxins	Storage Structure	Mean ± SE at Month			
		0	2	4	6
Aflatoxins (ppb)	PICS bags	4.35 ± 0.03	5.73 ± 0.06c	5.72 ± 0.02d	5.93 ± 0.03c
	SGP bags	4.33 ± 0.03	5.35 ± 0.03d	5.91 ± 0.02c	5.95 ± 0.03c
	PP bags	4.32 ± 0.01	6.91 ± 0.02a	12.91± 0.01b	15.73± 0.15b
	Jute bags	4.33 ± 0.02	5.95 ± 0.03b	14.04± 0.01a	16.14± 0.03a
Deoxynivalenol (ppm)	PICS bags	0.90 ± 0.00	0.92 ± 0.00c	0.92 ± 0.00c	0.92 ± 0.00c
	SGP bags	0.90 ± 0.00	0.92 ± 0.00c	0.92 ± 0.00c	0.93 ± 0.00c
	PP bags	0.90 ± 0.00	1.30 ± 0.00b	1.64 ± 0.00b	1.85 ± 0.01b
	Jute bags	0.90 ± 0.00	1.34 ± 0.00a	2.31 ± 0.00a	3.00 ± 0.00a
Fumonisins (ppm)	PICS bags	0.43 ± 0.00	0.45 ± 0.00c	0.45 ± 0.00c	0.45 ± 0.00c
	SGP bags	0.44 ± 0.00	0.44 ± 0.00c	0.44 ± 0.00c	0.45 ± 0.00c
	PP bags	0.43 ± 0.00	1.26 ± 0.00b	1.86 ± 0.00b	2.00 ± 0.00b
	Jute bags	0.43 ± 0.00	1.60 ± 0.01a	1.86 ± 0.00a	2.12 ± 0.00a
Ochratoxin A (ppb)	PICS bags	4.07 ± 0.11	4.63 ± 0.15c	5.70 ± 0.09c	5.16 ± 0.18c
	SGP bags	4.18 ± 0.28	4.29 ± 0.03d	5.04 ± 0.03d	5.37 ± 0.08c
	PP bags	4.33 ± 0.24	7.13 ± 0.03b	9.30 ± 0.17b	9.82 ± 0.20b
	Jute bags	4.29 ± 0.45	10.00 ± 0.18a	12.30± 0.35a	13.90± 0.14a

Each mean is based on n = 3 replications. SE that is 0.0 falls in the range of 0.0 to 0.003. Means for each mycotoxin among storage structures by month followed by different letters are significantly different (*p* < 0.05; by Tukey’s HSD test).

**Table 5 toxins-15-00039-t005:** Pearson’s correlation between fungal population and mycotoxins.

Storage Structures	Aflatoxins	Deoxynivalenol	Fumonisins	Ochratoxin
*Aspergillus*	0.94 ***	0.88 ***	0.96 ***	0.93 ***
*Fusarium*	−0.81 ***	−0.61 ***	−0.51 **	−0.41
*Penicillium*	0.96 ***	0.90 ***	0.94 ***	0.92 ***

Correlations are significant at ‘***’ 0.001 and ‘**’ 0.01 levels.

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
