# Peer review of "Effects of Storage Duration and Structures on Sesame Seed Germination, Mold Growth, and Mycotoxin Accumulation"

_toxins, 2023, doi:10.3390/toxins15010039_

Round 1

Reviewer 1 Report

This paper investigated the germination rate, mold growth and mycotoxin accumulation of sesame seeds in two hermetic bags (PICS bags and SGP bags) and two non-hermetic bags (PP bags and jute bags) under different storage conditions. Although this paper seems interesting, it is not suitable for publication at present. My detailed comments are listed as follows:

  1. The contents of the ABSTRACT were not clear, and the basis of the thesis should be mentioned.  
  2. The KEYWORDS were inaccurate.
  3. The novelty should be clarified in the Also, the current research progress of sesame seeds preservation methods was not fully explained. .
  4. There were many obvious errors in the RESULTS, for example:
    1. Figure 1 and Figure 2 were not correctly quoted in this text, and the contents of Figure 1 and Figure 2 were duplicated with Table 1 (Line 104-136);
    2. The carts and tables in the text were out of order;
    3. The title of Table 1 was repeated (Line 134);
    4. Species of molds mentioned in section 2.3 should be identified;
    5. The logical relationships between section 2.3 and 2.4 should be more clear;
    6. Sections 2.5 to 2.9 should be merged;
    7. The “* * * *” icon was not described in the footnote of Table 7 (Line 285).
  5. It is better to have corresponding subheadings in DISCUSSION to develop the interpretations logically and effectively, and comparisons between “insecticides” and methods involved in this study should be included in the DISCUSSION
  6. The article should be written more carefully, for instance:
    1. “a” in “remain a key a challenge” was redundant (Line 44);
    2. The “Quantitative” in “Quantitative and quality” should be in the form of nouns (Line 58);
    3. The expression of “and as well as” in the text was wrong. (Line 85);  
    4. The expressions of “0, 2, 4 and 6 months” in the text were not uniform;
    5. The concentration unit should not be expressed in“ ppb” and “ppm”;
    6. Abbreviation should be given when it was not the first time appeared in the text, such as “polypropylene”(Line 308-330);
    7. There should be space between numbers and units in the text, such as “0month”;
    8. The font size was not uniform (Line 249-264);
    9. The numerical symbols of “CO2 and O2” in the text should be expressed as subscripts.
    10. The formats of the references were not uniform, and should be carefully checked and corrected.

Author Response

I am/we are grateful to the reviewers for their insightful comments on my paper. I/We have been able to incorporate changes to reflect most of the suggestions provided by the reviewers. I have revised the manuscript according to reviewers' comments and tried to answer reviewers' comments point to point.

Reviewer 2 Report

  1. Remove from the title a double quotation marks.
  2. Latin names of fungi, plants and insects should be written in italics – line 49, 165, 167, 170, 172, 175, 177, 179, 182, 184, 319, 503, 507, 529, 548, 570, 578, 583, 587 and 594.
  3. Revise keywords, maybe instead ‘insecticides’ is better to use “pesticides”. The same applies to abstract line 20.
  4. Correct a sign of degree Celsius (now is 0 or º) – line 132, 134 (headline of tab. 1), in table 1 and 455.
  5. In table 4. correct name of fungal genera Penicillium now is Penicillin.
  6. The tables are wrongly numbered, after table 2 the next one is 4, then 6. There are also erroneous references to tables in the text, e.g. to non-existent tables 3 and 2.4.
  7. There are errors in the description of the results. Line 156 states "seed germination percentage decreased 90% to 61.7%" but should be 74% instead of 61.7%. Line157 instead 61.7% should be 71.5%. Line 278 instead r=0.87 should be r=0.86.
  8. Figure 6 is unnecessary the same results are in Table 6.
  9. Ahmad and Singh 1991 lines 338 and 426 - missing in references.
  10. SAS Institute 2013 line 487 – in text should be written [30].
  11. Chapter 5.5 Seed germination - It was reported that germination was performed according to Atlaf et al. 2004 (23) but in this publication germination test was performed using rolled paper method not top of paper. According to ISTA germination of sesame seeds is performed top of paper, in darkness at alternating temperature 20/30o Seedling are checked after 6 days. Please cite proper references. When evaluating germination capacity, were only normal germinated seedlings evaluated or all germinated seedlings? Are infected seedlings and dead seeds counted separately or together? These are two important germination parameters. In the table 2 'mold infection' is means the percentage of infected seedlings and dead seeds or the total number of seeds infected by fungi in the PDA test?
  12. Chapter 5.4 ‘Mold infection’ – lines 433-435 It was reported that slants were made to assess fungal species but only fungal genera were given in the results.
  13. Discussion and results – fungi belonging to genera Aspergillus and Penicillium are saprotrophic and storage fungi and they don’t infect seeds (they are not pathogenic fungi), they infest seeds. Please replace ‘infected” with “infested’.
  14. Results line 182-184 – Fusarium spp. are field fungi and during storage the percentage of infected seeds with these fungi are naturally decrease. I suggest you delete this sentence.
  15. References should be corrected in accordance with editorial requirements.

Author Response

(The authors gave the same response as above.)

Round 2
